**www.cambridge.org/qrd**

# Integrin force loading rate in mechanobiology: From model to molecular measurement

Hongyuan Zhang ⓘ, Micah Yang ⓘ, Seong Ho Kim and Isaac T.S. Li ⓘ

Department of Chemistry, The University of British Columbia, Kelowna, BC, Canada

## Perspective

DNA-based tension sensor; integrin; loading rate; mechanotransduction; molecular clutch; single-molecule force imaging

**Corresponding author:**
Isaac T. S. Li;
Email: isaac.li@ubc.ca

## Abstract

Integrins are critical transmembrane receptors that connect the extracellular matrix (ECM) to the intracellular cytoskeleton, playing a central role in mechanotransduction – the process by which cells convert mechanical stimuli into biochemical signals. The dynamic assembly and disassembly of integrin-mediated adhesions enable cells to adapt continuously to changing mechanical cues, regulating essential processes such as adhesion, migration, and proliferation. In this review, we explore the molecular clutch model as a framework for understanding the dynamics of integrin – ECM interactions, emphasizing the critical importance of force loading rate. We discuss how force loading rate bridges internal actomyosin-generated forces and ECM mechanical properties like stiffness and ligand density, determining whether sufficient force is transmitted to mechanosensitive proteins such as talin. This force transmission leads to talin unfolding and activation of downstream signalling pathways, ultimately influencing cellular responses. We also examine recent advances in single-molecule DNA tension sensors that have enabled direct measurements of integrin loading rates, refining the range to approximately 0.5–4 pN/s. These findings deepen our understanding of force-mediated mechanotransduction and underscore the need for improved sensor designs to overcome current limitations.

## Introduction

Cells are constantly exposed to various mechanical cues from their extracellular matrix (ECM) or neighbouring cells (Du *et al.*, 2023). Mechanotransduction is the fundamental process by which cells sense, integrate, and convert these physical stimuli into biochemical signals that regulate essential cellular functions (Du *et al.*, 2023; Huse, 2017; Zhang *et al.*, 2020). Among the key players in mechanotransduction are mechanosensitive molecules such as integrins (Pang *et al.*, 2023; Shen *et al.*, 2012), which serve as transmembrane receptors connecting the ECM to the intracellular actin cytoskeleton (Li *et al.*, 2016). The integrin family of cell adhesion receptors mediates bidirectional signalling between cells and their surroundings through 'inside-out' and 'outside-in' pathways. On the one hand, cells actively exert internal actomyosin cytoskeleton forces through integrins to activate integrin binding and deform their surroundings.

On the other hand, ligand binding to integrins transmits external forces from the ECM back to the cell, depending on ECM characteristics such as rigidity (Yi *et al.*, 2021), viscosity (Bennett *et al.*, 2018), and ligand spacing (Cavalcanti-Adam *et al.*, 2007). This bidirectional interaction ultimately influences cellular responses, including cell spreading, retraction, migration, and proliferation, while allowing cells to sense and adapt to their environment. Because it is constantly subjected to the force transmitted between cells and ECM, integrin acts as an ideal biomechanical sensor. Force experienced by integrin mechanically regulates its properties, including ligand-binding kinetics, conformation and activation, clustering and diffusion (Ali *et al.*, 2011; Chen *et al.*, 2017; Kechagia *et al.*, 2019). Upon binding to ECM components like fibronectin and collagen, integrins undergo conformational changes to be activated and cluster at the cell membrane. Following integrin clustering, adaptor proteins such as talin, vinculin, and paxillin are recruited to the adhesion sites to strengthen the integrin – ECM linkage, thus facilitating the formation of focal adhesions. These macromolecular assemblies anchor cells to the ECM and act as signalling hubs (Bauer *et al.*, 2019). Focal adhesion kinase and Src are key downstream nonreceptor tyrosine kinases of the formation of focal adhesions. They play a pivotal role in transducing signals from integrins to activate a range of signalling pathways, including the Ras-MAPK and PI3K-Akt pathways, which regulate cellular behaviours such as migration, proliferation, and survival (Bolós *et al.*, 2010; Westhoff *et al.*, 2004).

Integrin-mediated mechanosensitivity plays a critical role in various biological processes where cells sense and respond to mechanical cues from the ECM (Di *et al.*, 2023). First, integrin mediates tissue regeneration and wound healing (Kechagia *et al.*, 2019). Connective tissue repair involves fibroblasts, keratinocytes and endothelial cells (Koivisto *et al.*, 2014), which express a repertoire of integrins to sense and interact with the ECM. This interaction enables them to migrate toward the wound site and initiate directed migration, re-epithelization, granulation tissue formation, and wound contraction. Integrin is also essential for morphogenesis during

embryonic development (Molè *et al.*, 2021). As embryos develop, cells are sensitive to the mechanical properties of their surroundings. The interaction between integrins and various ECM components dictates the shape and adhesion pattern of stem cells, guiding their differentiation into specific lineages such as muscle, neural, or bone tissue (Estrach *et al.*, 2024; Lv *et al.*, 2015; Yi *et al.*, 2021). Moreover, immune cell activation and migration depend on integrin-mediated mechanosensing (Du *et al.*, 2023). For example, substrate stiffness modulates a range of T-cell behaviours, including migration (Saitakis *et al.*, 2017), cytokine secretion (Yuan *et al.*, 2021) and cytotoxic function (Saitakis *et al.*, 2017; Wang *et al.*, 2022b). Finally, in fibrotic diseases, integrins play a role in excessive ECM deposition, where activated fibroblasts sense increased matrix stiffness, leading to further ECM production and progression of fibrosis (Pang *et al.*, 2023; Yang and Plotnikov, 2021). Thus, integrin mechanosensitivity is vital for maintaining homeostasis in healthy tissues and can drive pathological changes when dysregulated.

Understanding the mechanical mechanisms at the molecular level is crucial for deciphering these fundamental biological processes. This review highlights the importance of investigating the integrin force loading rate and its biological relevance. We will examine this concept using the well-established molecular clutch model. Finally, we will summarise several recently developed single-molecule techniques for measuring the dynamics of forces, specifically the force loading rates, and discuss current limitations and future aspects.

## Dynamics of cell adhesion and the molecular clutch model

### The dynamic nature of cell adhesion

Although focal adhesions are robust and stable anchorages, they are dynamic rather than static (Ivaska, 2012). Integrins undergo cycles of activation-adhesion and inactivation-detachment, leading to the continuous assembly and disassembly of focal adhesions. This constant remodelling allows cells to firmly attach to the ECM and pull themselves forward during migration by generating traction forces. Integrin-mediated cell adhesion is crucial for directed migration. Cells dynamically assess and sample ECM rigidity by

applying variable pulling forces, guiding the process of durotaxis (Plotnikov *et al.*, 2012). Real-time traction force microscopy has revealed that cells exhibit tugging traction dynamics in focal adhesions on soft ECMs while they display stable traction on rigid ECMs. Because cells continuously interact with and adapt to ever-changing mechanical cues in their surroundings, understanding cell behaviours in response to their environment within a dynamic context is crucial.

### The molecular clutch model

The concept of 'molecular clutch' was introduced by Mitchison and Kirschner (1988) to depict the dynamic linkage between the cytoskeleton and the ECM. Clutches were initially defined as the dynamic linkage between actin filaments and the ECM through focal adhesion proteins and integrins. This concept has evolved and is now used to interpret cellular responses to various mechanical factors within the ECM. Clutches are currently referred to as the dynamic linkage formed by complexes comprising integrins and adaptor proteins (see Figure 1) (del Rio *et al.*, 2009).

Talin is a primary adapter protein that couples integrins to the actin cytoskeleton. When force is transmitted to talin, it unfolds, exposing previously hidden vinculin binding sites. This unfolding allows another adaptor protein, vinculin, to bind to talin with high affinity, further stabilising the integrin-actin linkage (Atherton *et al.*, 2015). In this framework, cells continuously generate forces via myosin, causing contraction of actin filaments and resulting in retrograde actin flow from the cell edge toward the centre. When integrins bind to extracellular substrates and couple the actin flow to the ECM, the clutch system engages. As a result, the retrograde flow pulls on the substrate, applying forces and potentially deforming it. Simultaneously, the elastic resistance of the substrate counters myosin contractility, slowing down the retrograde flow and increasing the force loading rate on the clutches (del Rio *et al.*, 2009). As force accumulates on talin up to a threshold level, talin unfolds, exposing vinculin binding sites and relieving vinculin's autoinhibition. Vinculin then binds to talin, strengthening the linkage between integrins and the actin cytoskeleton (Wang *et al.*, 2021; Yao *et al.*, 2014). The interaction between vinculin and the talin-integrin complex drives focal adhesion growth and integrin

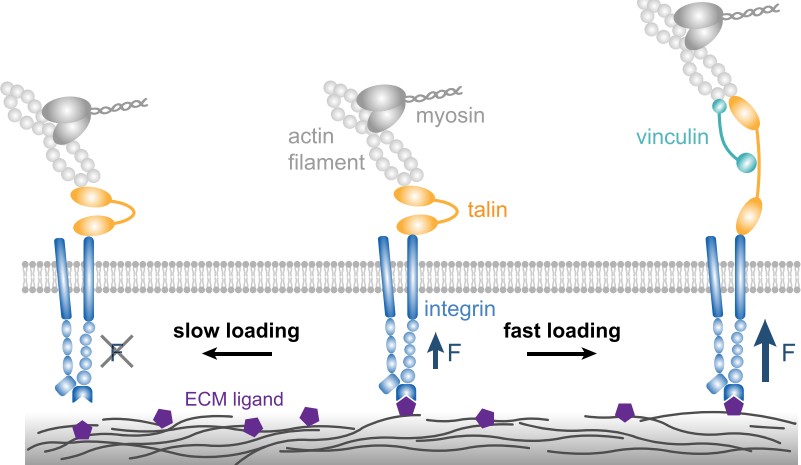

**Figure 1.** Schematic of molecular clutch model. The clutch represents the dynamic linkage between integrin and the ECM, mediated by adaptor proteins such as talin. Under fast force loading, the force accumulates beyond the threshold required for talin unfolding before the integrin – ECM bond disengages, thereby exposing vinculin binding sites. Vinculin binding reinforces the linkage. In contrast, under slow force loading, the integrin – ECM bond disengages before the force threshold for talin unfolding is reached, preventing vinculin binding. The bond rupture abolishes force transmission.

clustering, stabilising force transmission (del Rio *et al.*, 2009; Humphries *et al.*, 2007). As more integrins are recruited to the adhesion sites, additional clutches engage. This reduces the force applied to each clutch, preventing the disengagement of the system due to excessive force loading (Elosegui-Artola *et al.*, 2018).

The integrin – ECM linkage exhibits a catch–slip behaviour, where the bond lifetime initially increases with applied force (catch phase) and then decreases as the force continues to increase (slip phase) (Chen *et al.*, 2017; Kong *et al.*, 2009). As the force increases, the bond lifetime increases; however, as the force continues to build up, the bond eventually fails and results in the disengagement of the clutch. In contrast, the unfolding behaviour of talin domains follows a Bell-like model, where the unfolding rate increases exponentially with applied force (Bell, 1978). To achieve effective mechanotransduction, the force applied to talin must be loaded at an optimal rate that allows talin to unfold within the stable period of the integrin – ECM bond (See Figure 1).

The force loading rate is a core component of the molecular clutch model (Elosegui-Artola *et al.*, 2018), linking cellular mechanosensing to both actively generated forces within the cell and the passive mechanical properties of the ECM (Jiang *et al.*, 2016). The internal cellular machinery generates the active forces, mainly through actomyosin contraction. The passive mechanical properties are represented by the effective spring constant ($k$) of the ECM. This model defines the loading rate as the product of $k$ and actomyosin pulling speed ($v$) (Jiang *et al.*, 2016). From the perspective of loading rate, the molecular clutch model depicts biphasic behaviour in response to the ECM stiffness (Swaminathan and Waterman, 2016). On soft substrates, the compliance of the ECM buffers the retrograde movement of actin filaments driven by myosin, slowing the rate at which tension builds on each engaged clutch. When the force is loaded slowly, the integrin – ECM bond is more likely to fail before substantial force is transmitted to talin. In contrast, on rigid substrates, the force is loaded faster, allowing significant force to be transmitted to talin. This rapid force loading leads to talin unfolding, exposing previously cryptic vinculin binding sites and triggering subsequent mechanotransduction pathways.

Thus, the force loading rate is critical in determining whether force transmission through engaged clutches leads to effective mechanotransduction or clutch disengagement. Understanding this rate is essential for comprehending how cells respond to varying ECM stiffness and elucidating the mechanisms underlying cellular processes like migration, differentiation, and tissue development.

## Techniques for molecular force measurement

Researchers have developed various techniques to measure the magnitude of cellular forces (Liu *et al.*, 2017). These techniques can be broadly classified into three types:

1. Macroscopic deformation: This category includes traction force microscopy and micro-post array detectors, which measure substrate deformations under mechanical forces exerted by cells. While useful, these methods are limited to nanonewton resolution.
2. Instrument-based force spectroscopy: techniques such as atomic force microscopy, optical tweezers, magnetic tweezers, and biomembrane force probes fall under this category. These techniques allow force measurements at the single-molecule level but are limited by low throughput and spatial resolution (Bustamante *et al.*, 2021).

3. Molecular tension sensors: this includes tension sensor modules (TSMods) (LaCroix *et al.*, 2018), DNA hairpin probes (Zhang *et al.*, 2014), and tension gauge tethers (TGTs) (Wang and Ha, 2013). These sensors achieve piconewton (pN) resolution with high throughput, providing force readouts through fluorescence signals such as Förster resonance energy transfer (FRET) or fluorescence quenching.

The details of these three types of techniques, including their advantages and disadvantages, were extensively covered in the following excellent reviews (Fischer *et al.*, 2021; Liu *et al.*, 2017; Tu and Wang, 2020), hence we will not discuss them in further details here. We will primarily elaborate on molecular tension sensors. Genetically encoded TSMod incorporates proteins of interest into an elastic FRET module – a flexible peptide linker inserted between two fluorophores. When tension is applied to the protein, the elastic linker extends, decreasing FRET or quenching efficiency. The vinculin tension sensor (VinTS) is specifically designed to measure mechanical forces exerted on vinculin at focal adhesions (see Figure 2a) (Ayad *et al.*, 2022; Grashoff *et al.*, 2010). It consists of the head and tail domains of vinculin connected by a 40 amino acid (aa)-long elastomer domain. After calibration, VinTS can reliably report forces within the 1–6 pN range, with average forces across vinculin detected at approximately 2.5 pN (Grashoff *et al.*, 2010).

Unlike TSMod, which measures intracellular tension directly within the cell, DNA hairpin probes and TGT are typically coated onto substrates like glass coverslips to measure forces transmitted to transmembrane proteins from the extracellular environment. As its name suggests, the DNA hairpin probe consists of a single-stranded DNA sequence that folds back on itself to form a hairpin loop structure (see Figure 2b) (Zhang *et al.*, 2014). The end of the

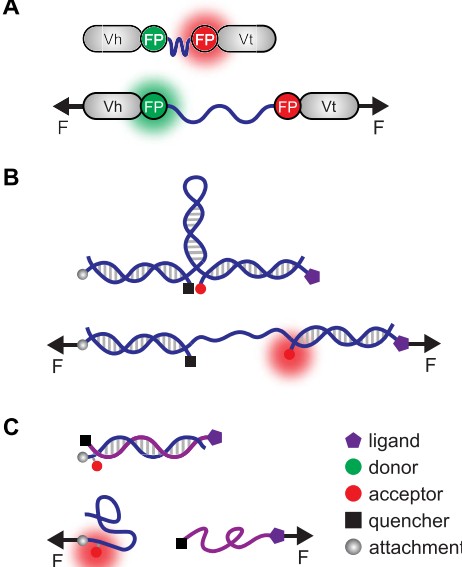

**Figure 2.** Schematic representations of various molecular force sensors. (a) VinTS comprising head (Vh) and tail (Vt) domains connected by an elastomeric peptide (blue) and a fluorescent protein (FP) FRET pair (red and green), with FRET signal decreasing upon peptide extension under tension; (b) DNA hairpin probe, where a fluorophore is quenched in the absence of tension but becomes fluorescent when the hairpin opens under sufficient tension, increasing the distance from the fluorophore to the quencher beyond its quenching range; (c) TGT, where a DNA duplex remains quenched when intact, and fluorescence occurs upon dissociation of the strand attached to a ligand (purple) from the surface-bound strand (blue) under applied tension.

hairpin is bioconjugated with a specific recognition motif, allowing cells to bind and interact with the sensor. When a cell exerts tension on the hairpin, the stem unfolds, separating the fluorophore and quencher. Due to its reversible folding and unfolding in response to mechanical forces, the DNA hairpin probe can monitor real-time tension forces and capture temporal oscillations of integrin tension force (Zhang *et al.*, 2014). These sensors can detect forces as low as 4.7 pN up to about 19 pN, tunable by sequence.

TGTs consist of double-stranded DNA modified to bind to cells and measure mechanical forces through fluorescence (see Figure 2c) (Wang *et al.*, 2015; Wang and Ha, 2013). TGTs record irreversible rupture events when cells produce sufficient tension to rupture them. The tension tolerance ($T_{tol}$), a metric describing the strength to resist mechanical rupture in TGT, is defined as 'the lowest force that ruptures the DNA within 2 seconds if the force is applied at a constant level' (Wang and Ha, 2013). Using TGT, researchers have revealed a close interplay between the magnitude of force and mechanotransduction. The integrin tension forces in CHO-K1 cells were reported to be able to rupture TGT with $T_{tol}$ ranging from 12 to 56 pN (Wang and Wang, 2016). The growth of focal adhesions correlates positively with integrin tension (Chang Chien *et al.*, 2022; Wang *et al.*, 2015). Specifically, the sizes of focal adhesions increased from 1 to 6 μm as cells were seeded onto TGT surfaces with increasing tension tolerances ($T_{tol}$ = 43–56 pN). Additionally, the translocation of yes-associated protein (YAP), a mechanosensitive transcription factor, from the cytoplasm to the nucleus occurs only when forces across integrins are steadily transmitted on higher $T_{tol}$ TGT ($T_{tol}$ = 50–54 pN).

It is important to note that cellular forces quantified by the molecular tension sensors require careful interpretation. The magnitude of the force transmitted by cells is greatly impacted by the mechanical properties of ECM (Humphrey *et al.*, 2014). For example, it has been reported that T cells can engage T-cell receptors (TCRs) on hard coverslips with forces sufficient to rupture TGTs with $T_{tol}$ = 12–19 pN (Liu *et al.*, 2016). However, on gel-phase supported lipid bilayers (SLBs), the rupture force imposed by TCR was approximately 5 pN (Göhring *et al.*, 2021). On the fluid-phase SLBs, the force was further reduced to 1.9 pN.

Furthermore, the reported $T_{tol}$ of TGTs cannot be directly interpreted as the actual force magnitude exerted by cells. Physiologically, cells likely apply forces over longer durations and dynamically in response to various environments (Gardel *et al.*, 2010; Gjorevski *et al.*, 2015), while $T_{tol}$ is calibrated within 2 seconds at a constant loading rate. Similarly, the value of $F_{1/2}$ of DNA hairpin probes requires careful calibration to reduce folding/unfolding hysteresis to report more accurately the dynamic and variable force loading experienced by cells in physiological environments (Yasunaga *et al.*, 2019).

Despite advancements in the development of first-generation molecular tension sensors, these tools often suffer from limited dynamic ranges or provide only binary outputs, indicating whether a specific force threshold has been exceeded. Such limitations make it challenging to accurately measure the dynamics of molecular tension, particularly the loading rate.

## Measuring molecular loading rate

Focusing solely on force magnitude overlooks the dynamic nature of cellular responses and the complexity of ECM mechanics. The concept of force loading rate fills this gap by accounting for how quickly the force is applied to molecular bonds, which directly influences whether bonds like integrin–ECM linkages can transmit sufficient force to mechanosensitive proteins before disengaging. This understanding is crucial for deciphering cellular behaviours responding to different mechanical environments.

### *Rupture force and bond lifetime depend on the loading rate*

The magnitude of the force exerted by cells is a critical parameter in mechanotransduction. However, focusing solely on force magnitude overlooks the dynamic nature of cellular responses to mechanical stimuli and the complexity of ECM mechanics. The concept of force loading rate fills this gap in understanding dynamic cell behaviours. It deciphers the complex ECM mechanics and translates mechanical signals into biochemical signals to mediate subsequent cellular responses. For instance, integrins have a lower loading rate on soft substrates than stiffer substrates, leading to lower integrin rupture force (Jiang *et al.*, 2016). It has long been recognized that force loading rate plays a significant role in molecular adhesion events like bond lifetime and rupture forces, thereby regulating related mechanosensing (Andreu *et al.*, 2021). Different loading rates can dramatically change the rupture forces of adhesion proteins, either abolishing or promoting mechanotransduction across the same set of protein–ligand interactions (Huang *et al.*, 2017; Liu *et al.*, 2014a; Ma *et al.*, 2022). This change can be exaggerated depending on the shape of the force-dependent lifetime curve of the bond in question.

Slip bonds, which decrease in lifetime with tension, remain stable at low force but break more readily at high forces. Thus, a slip bond experiencing a particular loading rate will sustain tension initially, with rupture probability increasing as force increases. In this case, a slower loading rate decreases the most probable rupture force; more time spent at a lower force increases the probability of rupture occurring at that force.

The effect is far more dramatic for catch bonds, which have a region where bond lifetime increases with force. A catch bond has a short lifetime at low forces, so at sufficiently slow loading rates, it cannot maintain tension. The loading rate must be fast enough to reach a stabilizing force before the catch bond ruptures. Several adhesive or mechanosensitive proteins, such as certain integrins (Chen *et al.*, 2010; Kong *et al.*, 2009), cadherins (Manibog *et al.*, 2014; Rakshit *et al.*, 2012), selectins (Barkan and Bruinsma, 2024; Evans *et al.*, 2004), actin (Guo and Guilford, 2006; Huang *et al.*, 2017), actin-binding domain of talin (Owen *et al.*, 2022), and TCRs (Liu *et al.*, 2014a; Ma *et al.*, 2022) have been found to exhibit catch-bond behaviour. Therefore, loading rate, in addition to force magnitude, is critical for a complete understanding of mechanotransduction.

### *Force loading rate bridges ECM mechanics to mechanotransduction*

While numerous studies have explored the role of matrix stiffness in mediating stem cell behaviour (Chen *et al.*, 2010; Manibog *et al.*, 2014; Rakshit *et al.*, 2012), much less is known about the mechanism by which matrix stiffness leads to changes in cell morphology, adhesion, proliferation and differentiation. Considering that the loading rate is the product of the effective spring constant of the ECM and the actomyosin pulling speed, changes in mechanical properties significantly affect the loading rate applied by cells and thus influence subsequent cellular behaviour (Jiang *et al.*, 2016). Force loading rate plays a vital role in translating substrate rigidity into intracellular signalling to regulate cell differentiation.

Mesenchymal stem cells tend to differentiate into neurogenic lineages on soft substrate, whereas they differentiate into osteogenic (bone) lineages on stiff substrate (Wang *et al.*, 2022a). Soft substrate limits the force cells apply to the substrate, thus modulating subsequent transcriptional activities. Mesenchymal stem cells on soft substrates exhibit less maturation of focal adhesions, reduced F-actin assembling, and more relaxed nuclei. Andreu *et al.* (2021) showed that the loading rate is a driving parameter of mechanosensing. They manipulated the loading rate by changing the substrate stiffness or the external stretching frequency. Their results demonstrated that increasing the loading rate leads to two major mechanosensitive events: talin-mediated adhesion growth and reinforcement and YAP translocation from cytosol to the nucleus.

A higher force loading rate ensures the force is transmitted to talin and induces its unfolding before the integrin – ligand bond disengages. When talin unfolds, it exposes binding sites for vinculin, which strengthens the connection between talin and F-actin, enhancing force transmission by recruiting additional actin filaments (Li *et al.*, 2016). The forces generated at focal adhesions can be transmitted to the nucleus, stretching nuclear pores and facilitating the entry of YAP into the nucleus (Elosegui-Artola *et al.*, 2017). Once inside, YAP interacts with TEA domain (TEAD) transcription factors to regulate gene expression. The YAP-TEAD complex promotes cell proliferation and inhibits apoptosis by controlling the expression of target genes (Kwon *et al.*, 2022).

In the context of osteogenesis, YAP plays a complex role alongside the transcriptional coactivator with PDZ-binding motif (TAZ) (Pan *et al.*, 2018; Wang *et al.*, 2023a). TAZ actively promotes osteogenesis by coactivating runt-related transcription factor 2 (RUNX2) genes, which are critical for bone development. On the other hand, YAP has a dual role: it can inhibit RUNX2-mediated transcription, thereby downregulating osteogenesis while

stabilizing β-catenin to enhance β-catenin-mediated osteogenesis (Pan *et al.*, 2018).

In summary, ECM mechanics, such as stiffness, regulate the force-loading rate onto the cell via integrin. The loading rate determines whether sufficient force can be transmitted to critical mechanosensitive proteins like talin, leading to their activation and triggering downstream signalling pathways and cell behaviours before the integrin – ECM linkage disengages.

### Methods to quantify integrin loading rate

While molecular tension sensors allow quantification of force magnitude at the pN level, they do not measure the loading rate of integrin tension. Moore and colleagues estimated the force loading rate of a single integrin by measuring the deformation of the elastomeric substrate, reporting values from 0.007 to 4 pN/s (Moore *et al.*, 2010). While this method provided rough estimation, direct measurements at the single-molecule level were needed. In light of this deficiency, three groups recently developed dual DNA tension sensors that directly reported force loading rates at the single-molecule level.

The Ha group developed an overstretching tension sensor (OTS) based on stretching-induced oligonucleotide dehybridization (see Figure 3a) (Jo *et al.*, 2024). They connected two OTSs with distinct dehybridization forces of 16 and 30 pN, labelled with different fluorophores (Atto674N and Cy3). By recording the time interval between the two fluorescence signals when each threshold force was reached, they calculated the loading rate as the force difference divided by this time interval. Using OTSs, they reported that the integrin loading rate ranged from 0.5 to 4 pN/s.

The Salita group developed a loading rate probe (LR probe) that incorporated two oligonucleotide strands, each of which undergoes

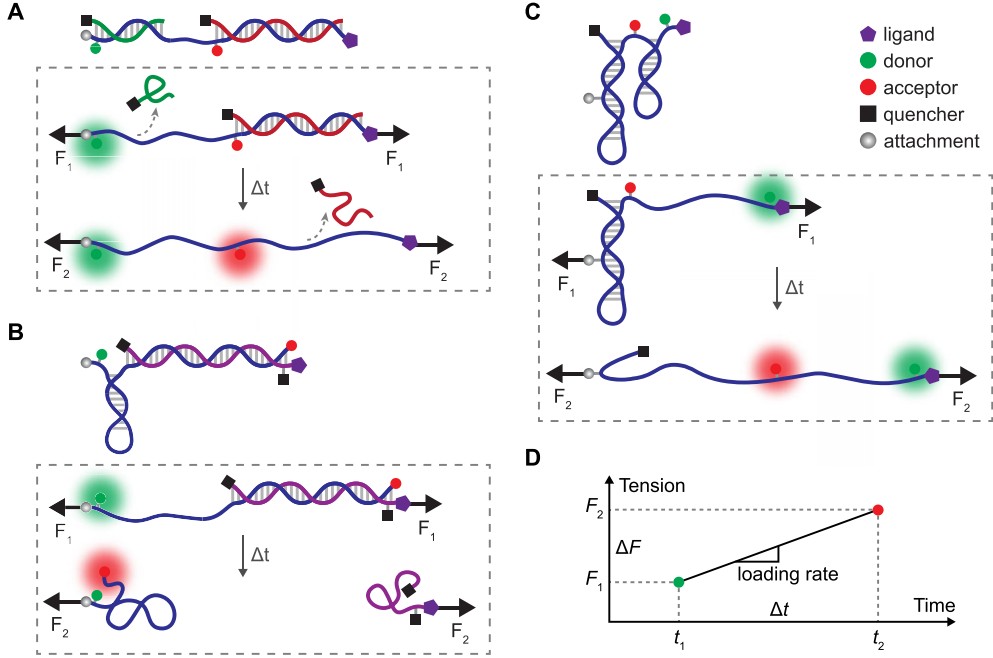

**Figure 3.** Schematic of three recently developed force-loading rate sensors. (a) OTS, where forces exceeding $F_1$ and $F_2$ sequentially displace two DNA duplexes (green and red), unquenching their corresponding fluorescence signals (green and red) in order; (b) LR probe, consisting of a DNA hairpin that opens at force $F_1$, connected to a TGT designed to rupture at a higher force $F_2$, detecting two sequential events, with the final event causing the surface attached DNA to recoil and a high-FRET (red) signal; (c) ForceChrono probe, utilizing two DNA hairpins with distinct attachment geometries that open sequentially as force increases from $F_1$ to $F_2$, resulting in the sequential appearance of red and green fluorescence signals. (d) Given the designed force difference ($\Delta F$) and time difference ($\Delta t$) between the two events, the loading rate can be determined, assuming linear force ramp between the two events.

a conformational change at different force thresholds and reports unique fluorescence signals (see Figure 3b) (Combs *et al.*, 2024). A lower force threshold at 4.7 pN leads to hairpin unfolding, and as force increases, a duplex TGT (with a $T_{tol}$ of 56 pN) gets sheared. The results showed the median loading rate of integrin-mediated force as 1.3 pN/s.

The Liu group designed a ForceChrono probe consisting of two DNA hairpins labelled with distinct fluorophores, each unfolding at different force thresholds (Hu *et al.*, 2024). They developed two versions of ForceChrono probes to cover broader mechanical ranges, one for 7–19 pN and another for 17–41 pN forces (see Figure 3c). The average loading rates derived from these two ForceChrono probes were 0.6 and 1.5 pN/s, respectively. Their single-molecule trajectories revealed a spatio-temporal heterogeneity in the dynamics of integrins where the integrin – talin – actin linkages are initially (first 20 minutes) unstable with faster loading rates (~0.9 pN/s) and shorter force durations (~45 s). After 8 hours, as focal adhesions stabilized, the loading rate decreased (~0.5 pN/s), and force duration increased (~100 s). This feature was consistent with the previously discussed cell dynamics observed by traction force microscopy, where cells showed tugging traction force on a soft substrate but exhibited stable traction force on a rigid substrate (Plotnikov *et al.*, 2012).

Collectively, the measured loading rates in these three studies overlapped significantly, and the researchers managed to refine this measurement to a much more precise range.

## Consideration, challenges, and future perspectives

### Effects of substrate rigidity on loading rate

Rigidity is an essential characteristic of ECM properties. Physiological rigidity varies significantly across tissues – from soft brain tissue (1–4 kPa) to stiff bone tissue (1000–1500 kPa) (Handorf *et al.*, 2015). While current studies are performed on hard coverslips to quantify in vivo integrin loading rates (Combs *et al.*, 2024; Hu *et al.*, 2024; Jo *et al.*, 2024), these coverslips are much stiffer than tissues. This could potentially take advantage of the method from Hu and colleagues. They were able to monitor molecular tension at different substrate stiffness by coating DNA tension sensors on soft hydrogels (Wang *et al.*, 2023b). They fabricated a series of hydrogels with different moduli ranging from 1 to 80 kPa and coated DNA tension sensors on the soft surface through golden nanoparticles. Their results demonstrated that cells recruit more force-bearing integrins and adjust their interaction dynamics with the ECM to form stronger, more mature focal adhesions on rigid substrates, which is consistent with what the molecular clutch model suggests (Elosegui-Artola *et al.*, 2018). Combining this methodology with some advancement in single-molecule imaging in 3D would be very interesting to see how the substrate stiffness alters the loading rate on integrins.

### Influence of ligand density on loading rate

Ligand density is also a crucial factor in the ECM environment, affecting cellular adhesion structures and force-mediated mechanosensing (Liu *et al.*, 2014b; Oria *et al.*, 2017). Schvartzman and colleagues demonstrated a significant increase in cell spreading efficiency when clusters of at least 4 liganded integrins were within approximately 60 nm – a spacing within physiological ranges of 10–200 nm (Le Saux *et al.*, 2011; Schvartzman *et al.*, 2011). Considering force balance at the interface, ligand spacing plays a significant role

in measuring the loading rate *in vivo*. As integrin binds to ligands to engage the clutch system, the force transmitted to ECM counters myosin contractility, thereby decreasing actomyosin pulling speed (*v*) (Barnhart *et al.*, 2011; Elosegui-Artola *et al.*, 2018). Given a constant and optimal rigidity, increasing ligand density increases the number of clutches engaged, thereby slowing down the pulling speed and resulting in a lowered loading rate, which is the product of the effective spring constant of the substrate (*k*) and actomyosin pulling speed (*v*). Hu and colleagues investigated the impact of ligand density on integrin loading rates. They found that at lower ligand spacing (40 nm), the average loading rate was slower (~0.3 pN/s) and force duration longer (~180 s) compared to higher ligand spacing (100 nm), where the loading rate was faster (~1.25 pN/s) with shorter force duration (~90 s). These results were consistent with the molecular clutch model: higher ligand density allows force to be more stably exerted and distributed over more adhesion points, strengthening integrin – talin – actin linkages. Conversely, lower ligand density leads to less stable force distribution, resulting in instability and frequent bond ruptures. Given there are differences due to integrin density, a systematic investigation of how this affects the loading rate could shed light on the different biological processes that can be controlled entirely by the ligand density.

### Interpreting readout from loading rate sensor

While current molecular tension sensors have provided initial insights into the force-loading rates of integrins, there is significant room for improvement. Current techniques for measuring integrin loading rates possess inherent observation biases that must be carefully considered during data interpretation.

All current techniques rely on the sequential detection of two fluorescent events: the first occurs at $t_1$, indicating the opening of DNA duplex $d_1$ at force $F_1$; the second occurs at $t_2$, indicating the opening of DNA duplex $d_2$ at force $F_2$. The sequence of these events is crucial because $F_1$ is designed to be lower than $F_2$. Thus, the only data traces that contain both signals in the correct order are interpretable.

This reliance introduces the first bias that events that do not reach $F_1$ are undetected, and events that do not reach $F_2$ are discarded (Figure 4b). This introduces a bias of only representing the loading rates of events that ultimately reached sufficiently high tension. This limitation is particularly problematic when measuring catch bonds (Figure 4b), which many mechanosensitive receptors are. Catch bonds have a characteristic double rupture force distribution. The higher force rupture peak is dominant at a high loading rate, but at a low loading rate, the low rupture force events dominate. Due to this, catch bonds with a slow loading rate may not be observed, meaning a potentially large subset of functionally important behaviours is underrepresented if not entirely missing. Therefore, the nature of the adhesion interactions (i.e., catch vs. slip) must be considered when designing the loading rate sensor.

Furthermore, interpreting the data involves assuming a constant loading rate between $t_1$ and $t_2$ within the force range between $F_1$ and $F_2$. This assumption rests on two key premises: (1) the force difference ($\Delta F$) between $F_1$ and $F_2$ remains constant, and (2) that force loading is constant over the time interval ($\Delta t$) (Figure 3d, 4a). The first assumption must be carefully designed or accounted for in subsequent analysis because DNA nanomechanics are sensitive to temperature, salt concentration, molecular crowding, and force loading rate. A well-designed loading rate sensor should utilize $d_1$ and $d_2$ duplexes that are either equally affected by or insensitive to these

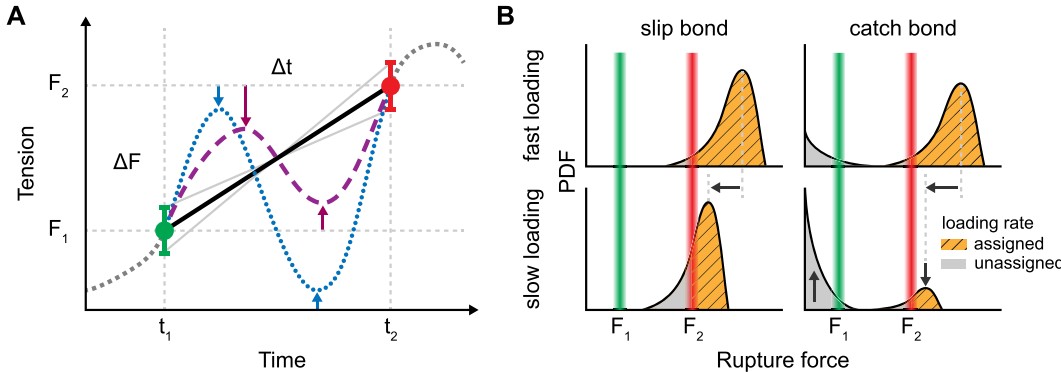

**Figure 4.** Potential challenges in interpreting data from current loading rate sensors. (s) Due to the stochastic nature of bond rupture, rupture forces have distributions around $F_1$ and $F_2$ (illustrated by error bars) and may be dependent on the loading rate, introducing potential inaccuracies in the assumed linear loading rate. Additionally, different force trajectories (blue dotted line and purple dashed line) can produce identical observed signals. In reversible sensors (purple dashed line) that emit a green signal at $F_1$, the force range is confined between $F_1$ and $F_2$. In contrast, for irreversible sensors (blue dotted line) generating a green signal, the force is only constrained by an upper bound at $F_2$, while it can decrease toward zero before rising again to $F_2$ to produce a red signal. As a result, assuming a linear force ramp may be an oversimplification, especially if the duration of events is long. (b) The nature of catch or slip bonds under varying loading rates can obscure certain events. The graphs depict catch or slip behaviours at fast and slow loading rates. The green and red lines represent the sensor rupture forces at $F_1$ and $F_2$, respectively. The striped yellow and grey regions under the rupture force distributions represent the populations of native events where the loading rate can (striped yellow) and cannot (grey) be assigned. Receptor-ligand rupture events below $F_2$ cannot be assigned a loading rate, which biases loading rate observations toward events that occur above $F_2$. This is particularly problematic for catch bonds, where the bimodal distribution of rupture forces includes a low-force component that dominates at low loading rates.

factors – ensuring that $\Delta F$ remains constant even if the absolute values of $F_1$ and $F_2$ change (Hu *et al.*, 2024). This minimizes the impact of varying conditions on the loading rate measurement.

While the current designs have addressed the first assumption to some extent, the second assumption presents a greater challenge with current loading rate sensors. Because the sensors report discrete events, they inherently miss the force dynamics between $t_1$ and $t_2$. Therefore, the shorter the $\Delta t$, the more likely a linear approximation of force loading reflects the underlying reality. For longer $\Delta t$, the linear approximations become less accurate due to the time scale of tension dynamics (tens of seconds) (Puklin-Faucher and Sheetz, 2009) and the possibilities of many force trajectories that pass through both $F_2$ at $t_1$ and $F_2$ at $t_2$ (Figure 4a). One approach to improve the accuracy of data interpretation for loading rate sensors is to decrease $\Delta t$ or $\Delta F$, albeit at the expense of dynamic range, and multiplex these sensors to obtain a comprehensive picture of loading rates across a broader force range. Alternatively, increasing the number of discrete duplexes that rupture at different forces within the same construct can refine force detection.

Similarly, an analogue tension sensor with a large force dynamic range may achieve better temporal resolution. The design of loading rate sensors can also exclude behaviours which violate the second assumption: In the case of reversible constructs with minimal unfolding/refolding hysteresis, one can ensure that the force remains above $F_1$ while waiting to reach $F_2$, eliminating oscillating force trajectories, as well as unbinding/rebinding of different ligands. For irreversible constructs, there is no guarantee that the force remains above $F_1$ before $F_2$ appears. Current loading rate sensor designs cannot exclude force plateaus, leading to a potential underestimation of the loading rate; this is an opportunity for new, innovative designs moving forward.

## Conclusion

Accurately measuring the force loading rate is crucial for understanding how cells convert mechanical cues from their environment into biochemical signals that regulate vital functions. Recent advances in single-molecule tension sensor technology, particularly dual DNA tension sensors, have significantly enhanced our ability to measure integrin loading rates with high precision. Combining these advanced measurement techniques with systematic studies of ligand density and substrate stiffness while addressing current methods' limitations can further refine our understanding of integrin-mediated mechanotransduction and its role in cellular functions.

**Open peer review.** To view the open peer review materials for this article, please visit http://doi.org/10.1017/qrd.2024.28.

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
