## [Reviewer Report]

Li and coworkers, in this timely perspective, provide an informative overview of the biological and methodological developments leading to the current cell force sensor and now loading rate sensor toolbox. The authors astutely identify the key limitations of current designs and it is important to communicate these to the wider mechanobiology community before concrete biological conclusions are drawn from their implementation. There remain many open questions when drawing conclusions from force sensors that it is clear that loading rate sensors based on force sensors must be used cautiously.

I have some minor comments regarding readability of figures and text. Otherwise, I strongly support the publication of this manuscript in QRB Discovery after these concerns are addressed:

1. The figures of the manuscript are generally understandable. However, I question the use of red/green for illustration purposes without an additional colour free identifier (shape, shade, pattern and so on) due to colour blindness and how the figures present in grey scale. Figures 1 and 2 are possible to interpret for those knowledgeable about FRET. However, figure 3 poses a challenge to interpret due to the similarity of colours chosen and lack of identifiers. This figure suffers similar issues with blue/purple and yellow/grey mentioned in the figure legend but are not helpful unless one can directly observe the colour. I would recommend adjustment of the figures with some more non-colour identifiers so that they are more readily interpreted.

2. I noticed a general overuse of abbreviations that may confuse and interrupt readers. Consider removing some of the less needed abbreviations that are used only 1-2 times. For example, VBS for vinculin binding sites may not be absolutely required especially as later it is referred to as binding sites for vinculin.

3. I identified inconsistencies in naming conventions for the different probes. For example, chrono phore and Chrono phore, Vinculin Tension Sensor and vinculin tension sensor, overstretching tension sensor and Overstretching Tension Sensor, and so on. This should be unified to prevent mental load for the reader.

---

## [Reviewer Report]

The authors elaborated really well in introducing a variety of techniques to study proteins involved in mechanobiology. Few minor points,

1. The title “Force loading rate in mechanobiology: from model to molecular measurement” seems to be too broad since the review is mostly focus on integrins/talin.

2. In the paragraph starting in line 111, the review discusses about the clutch model being important to understand mechanotransduction. It would be very useful if the authors add a cartoon of the this clutch and catch-slip model to illustrate.

3. The authors should mention the recent experimentally reported catch-bind behavior of the talin-actin interactions using dual trap optical tweezers by the Alexander R. Dunn at Standford: “The C-terminal actin-binding domain of talin forms an asymmetric catch bond with F-actin”. Maybe this reference can be incorporated in the paragraph between lines 249 and 257.

4. The reviewer is not familiar with the methods and literature of this specific field. Since a review tries to illustrate to a non-specialized scientific audience, I think it would be useful if the authors can add a table (maybe one or two), including the types of methods, the advantages, disadvantages, and references of studies using an specific method.